

# Mismatch between IUCN range maps and species interactions data illustrated using the Serengeti food web

Gracielle T. Higino[1], Francis Banville[2,3,4], Gabriel Dansereau[3,4], Norma Rocio Forero Muñoz[3,4], Fredric Windsor[5] and Timothée Poisot[3,4]

[1] Biodiversity Research Centre, University of British Columbia, Vancouver, British Columbia, Canada
[2] University of Sherbrooke, Sherbrooke, Québec, Canada
[3] University of Montreal, Montréal, Québec, Canada
[4] Quebec Centre for Biodiversity Science, Montréal, Québec, Canada
[5] School of Natural and Environmental Sciences, Newcastle University, Newcastle upon Tyne, United Kingdom

## ABSTRACT

**Background**. Range maps are a useful tool to describe the spatial distribution of species. However, they need to be used with caution, as they essentially represent a rough approximation of a species' suitable habitats. When stacked together, the resulting communities in each grid cell may not always be realistic, especially when species interactions are taken into account. Here we show the extent of the mismatch between range maps, provided by the International Union for Conservation of Nature (IUCN), and species interactions data. More precisely, we show that local networks built from those stacked range maps often yield unrealistic communities, where species of higher trophic levels are completely disconnected from primary producers.

**Methodology**. We used the well-described Serengeti food web of mammals and plants as our case study, and identify areas of data mismatch within predators' range maps by taking into account food web structure. We then used occurrence data from the Global Biodiversity Information Facility (GBIF) to investigate where data is most lacking.

**Results**. We found that most predator ranges comprised large areas without any overlapping distribution of their prey. However, many of these areas contained GBIF occurrences of the predator.

**Conclusions**. Our results suggest that the mismatch between both data sources could be due either to the lack of information about ecological interactions or the geographical occurrence of prey. We finally discuss general guidelines to help identify defective data among distributions and interactions data, and we recommend this method as a valuable way to assess whether the occurrence data that are being used, even if incomplete, are ecologically accurate.

Corresponding authors
Gracielle T. Higino, graciellehigino@gmail.com
Timothée Poisot, timothee.poisot@umontreal.ca

## INTRODUCTION

Finding a species in a certain location is like finding an encrypted message that travelled through time. It carries the species' evolutionary history, migration patterns, as well as any

direct and indirect effects generated by other species (some of which we may not even know exist). Ecologists have been trying to decode this message with progressively more powerful tools, from their field notes to highly complex computational algorithms. However, to succeed in this challenge it is important to have the right clues in hand. There are many ways we can be misled by data—or the lack of it: taxonomic errors (*e.g.,* due to updates in the taxonomy of a species), geographic inaccuracy (*e.g.,* approximate coordinates or lack of documentation about their accuracy), or sampling biases (*e.g.,* data clustered near roads or research centers) (*Ladle & Hortal, 2013*; *Hortal et al., 2015*; *Poisot et al., 2021*). One way to identify—and potentially fix—these errors is to combine many different pieces of information about the occurrence of a species, so agreements and mismatches can emerge. Although previous studies have combined different types of occurrence data to measure the accuracy of datasets (*Hurlbert & Jetz, 2007*; *Hurlbert & White, 2005*; *Ficetola et al., 2014*), none have used different types of information so far (*i.e.,* ecological characteristics other than geographical distribution). Here we suggest jointly analysing species occurrence (range maps and point occurrences) and ecological interactions to identify mismatches between datasets and areas of data deficit.

Interactions form complex networks that shape ecological structures and maintain the essential functions of ecosystems, such as seed dispersal, pollination, and biological control (*Albrecht, 2018*; *Fricke et al., 2022*) that ultimately affect the composition, richness, and successional patterns of communities across biomes. Yet, the connection between occurrence and interaction data is a frequent debate in ecology (*Blanchet, Cazelles & Gravel, 2020*; *Wisz et al., 2013*). For instance, macroecological models are often used with point or range occurrence data in order to investigate the dynamics of a species with its environment. However, these models do not account for ecological interactions, although it has been demonstrated that they might largely affect species distribution (*Abrego et al., 2021*; *Afkhami, McIntyre & Strauss, 2014*; *Araújo, Marcondes-Machado & Costa, 2014*; *Godsoe et al., 2017*; *Godsoe & Harmon, 2012*; *Gotelli, Graves & Rahbek, 2010*; *Wisz et al., 2013*). Some researchers argue that occurrence data can also capture real-time interactions (see *Roy, Saunders & Pocock, 2016*; *Ryan et al., 2018*), and, because of that, it would not be necessary to include ecological interaction dynamics in macroecological models. On the other hand, many mechanistic simulation models in ecology have considered the effect of competition and facilitation in range shifts. For example, *Gotelli, Graves & Rahbek (2010)* demonstrate how conspecific attraction might be the main factor driving the distribution of migratory birds; *Afkhami, McIntyre & Strauss (2014)* explores how mutualistic fungal endophytes are responsible for expanding the range of native grass; many other examples are discussed in *Wisz et al. (2013)*. Although interactions across trophic levels are demonstrated to determine species range (*Wisz et al., 2013*), the use of these interactions in mechanistic simulation models in macroecology remains insufficient (as discussed in *Cabral, Valente & Hartig, 2017*).

A significant challenge in this debate is the quality and quantity of species distribution and ecological data (*Boakes et al., 2010*; *Ronquillo et al., 2020*; *Meyer, Weigelt & Kreft, 2016*)—a gap that can lead to erroneous conclusions in macroecological research (*Hortal et al., 2008*). Amongst the geographical data available are the range maps provided by the

International Union for the Conservation of Nature (IUCN). Such maps consist of simplified polygons, often created as alpha or convex hulls around known species locations, refined by expert knowledge about the species (*IUCN Red List Technical Working Group, 2019*). These maps can be used in macroecological inferences in the lack of more precise information (*Fourcade, 2016*; *Alhajeri & Fourcade, 2019*), but it has been recommended that they are used with caution since they tend to underestimate the distribution of species that are not well-known (*Herkt, Skidmore & Fahr, 2017*) (especially at fine scale resolutions; Hurlbert and Jetz (2007); *Hurlbert & White (2005)*), do not represent spatial variation in species occurrence and abundance (*Dallas, Pironon & Santini, 2020*), and can include inadequate areas within the estimated range. Another source of species distribution information is the Global Biodiversity Information Facility (GBIF), which is an online repository of georeferenced observational records that come from various sources, including community science programs, museum collections, and long-term monitoring schemes. A great source of bias in these datasets is the irregular sampling effort, with more occurrences originating from attractive and accessible areas and observation of charismatic species (*Alhajeri & Fourcade, 2019*). As for ecological data, a complete assessment is difficult and is aggravated by biased sampling methods, data aggregation (*Poisot et al., 2020*; *Hortal et al., 2015*) and by the fact that interactions are very often events that occur in a narrow window of time. Nevertheless, we have witnessed an increase in the availability of biodiversity data in the last decades, including those collected through community science projects (*Callaghan et al., 2019*; *Pocock et al., 2015*) and dedicated databases, such as Mangal (*Poisot et al., 2016*). This provides an opportunity to merge species distribution and ecological interaction data to improve our predictions of where a species may be found across large spatial scales.

It has been demonstrated that the agreement between range maps and point data varies geographically (*Hurlbert & Jetz, 2007*; *Hurlbert & White, 2005*; *Ficetola et al., 2014*). Adding ecological interaction data to this comparison might help to elucidate where these (dis)agreements are more likely to be true and which dataset better represent the actual distribution of a species. In this context, we elaborate a method that allows us to detect areas of potential misestimation of species' distribution data (more precisely range maps) based on interaction data. This method is based on the assumption that organisms cannot persist in an area unless they are directly or indirectly connected to a primary producer within their associated food web (*Power, 1992*). Thus, given that herbivores are the main connection between plant resources (directly limited by environmental conditions) and predators (*Dobson, 2009*; *Scott et al., 2018*), the range of a predator (omnivore or carnivore) depends on the overlapping ranges of its herbivore preys. If sections of a predator's range do not overlap with at least one of its prey it will become disconnected from primary producers, and therefore we would not expect the predator to occur in this area.

This mismatch can be the result of different mechanisms, like the misestimation of both the predator's and the preys' ranges (*Ladle & Hortal, 2013*; *Rondinini et al., 2006*), taxonomic errors (*Isaac, Mallet & Mace, 2004*; *Ladle & Hortal, 2013*), or the lack of information about trophic links (*i.e.*, the lack of connection between the ranges of a predator and a primary producer may be due a third species we don't know is connected
to both). Here in this proof of concept, we investigate the disagreements between available data for species that compose a well-known food web in the African continent, discuss the mechanisms that can lead to this, and reinforce the importance of open geographically explicit interaction data.

## METHODS

We identified areas of data deficits within the ranges of predators based on a simple rule: any part of a predator's range that did not intersect with the range of at least one prey herbivore species, which in turn is directly connected to a primary producer (plants), was considered data deficient. To do that, we used a Serengeti food web dataset (*Baskerville et al., 2011*) (which comprises carnivores, herbivores, and plants from Tanzania) and its species ranges from IUCN. Then, we calculated the difference in range sizes between the original IUCN ranges of predators and those without the areas where they would be disconnected from their food webs, based on species interaction data. Finally, we added the GBIF occurrence points for the Serengeti species to investigate whether the results would be different if we used another source of distribution data.

### Data

We investigated the mismatch between savannah species ranges and interactions in Africa (Fig. 1). These ecosystems host a range of different species, including the well-characterized predator–prey dynamics between iconic predators (*e.g.*, lions, hyenas, and leopards) and large herbivores (*e.g.*, antelopes, wildebeests, and zebras), as well as a range of herbivorous and carnivorous small mammals. The Serengeti ecosystem has been extensively studied and its food web is one of the most complete we have to date, including primary producers identified to the species level. Here we focus on six groups of herbivores and carnivores from the Serengeti Food Web Data Set (*Baskerville et al., 2011*). These species exhibit direct antagonistic (predator–prey) interactions with one another and are commonly found across savannah ecosystems on the African continent (*McNaughton, 1992*). Plants in the network were included indirectly in our analyses as we do not expect the primary producers to significantly influence the range of herbivores for several reasons. Firstly, many savannah plants are functionally similar (*i.e.*, grasses, trees and shrubs) and cooccur across the same habitats (*Baskerville et al., 2011*). Secondly, herbivores in the network are broadly generalists feeding on a wide range of different plants across habitats. Indeed, out of 129 plants in our dataset, herbivores ($n = 23$) had a mean out degree (mean number of preys) of around 22 (std $= 17.5$). There is also an absence of global range maps for many plant species (*Daru, 2020*), which prevents their direct inclusion in our analysis. Therefore, we assume that plants consumed by herbivores are present across their ranges, and as such the ranges of herbivores are not expected to be significantly constrained by the availability of food plants.

From the wider ecological network presented in *Baskerville et al. (2011)*, we sampled interaction data for herbivores and carnivores. This subnetwork contained 32 taxa (23 herbivores and nine carnivores) and 84 interactions and had a connectance of 0.08. Although self-loops are informative, we removed these interactions to allow for the original

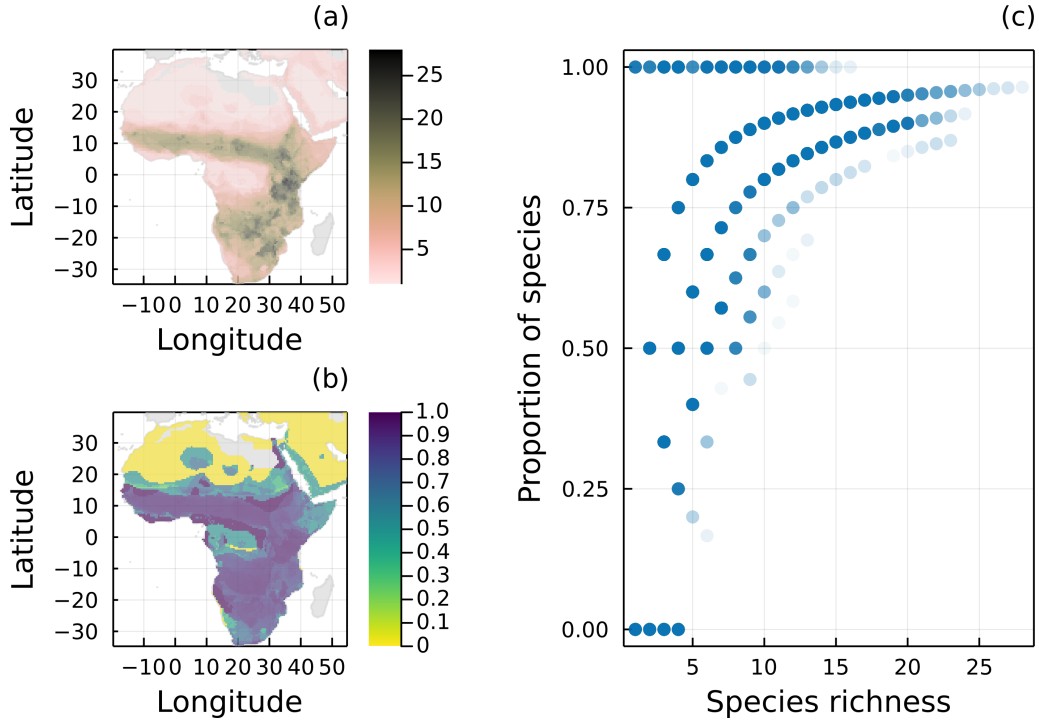

**Figure 1** **Geographical distribution of species richness and removal of predators.** (A) Spatial distribution of species richness according to the original IUCN range maps of all 32 mammal species of the Serengeti food web. (B) Proportion of mammal species remaining in each local network (*i.e.,* each pixel) after removing all species without a path to a primary producer. (C) Proportion of mammal species remaining in each local network as a function of the number of species given by the original IUCN range maps.

IUCN ranges of predators with cannibalistic interactions to be adjusted. We treated this overall network as a metaweb since it *should* contain all potential species interactions between mammalian taxa occurring across savannah ecosystems such as the Serengeti.

We compiled IUCN range maps for the 32 species included in the metaweb from the Spatial Data Download portal (http://www.iucnredlist.org/resources/spatial-data-download), which we rasterized at a 0.5 degrees resolution (∼50 km at the equator). We restricted the rasters to a spatial extent comprised between latitudes 35°S and 40°N and longitudes 20°W and 55°E. We then combined interaction data from the metaweb and cooccurrence data generated from species ranges to create networks for each raster pixel. This generated a total of 11,308 pixel-level networks. These networks describe potential predation, not actual interactions: the former is derived information from the metaweb, and the latter is contingent on the presence of herbivores.

## Range overlap measurement

We calculated the geographical overlap, *i.e.,* the extent to which interacting predator and prey species co-occurred across their ranges, as $a/(a+c)$, where $a$ is the number of pixels where predator and prey cooccur and $c$ is the number of pixels where only the focal species occur. This index of geographical overlap can be calculated with prey
or predators as the focal species. Values vary between 0 and 1, with values closer to 1 indicating that there is a large overlap in the ranges of the two species and values closer to 0 indicating low cooccurrence across their ranges. For each predator species, we calculated its generality to understand whether the level of trophic specialization (*i.e.,* number of prey items per predator) affects the extent to which the ranges of the species comprised areas of data deficits. One would assume that predators with a greater number of prey taxa (*i.e.,* a higher generality) are less likely to have large areas of data mismatch within their range as it is more likely that at least one prey species is present across most of their range.

## Validation

For each species in the dataset we collated point observation data from GBIF (http://www.gbif.org). We used the GBIF download API to retrieve all species occurrences on November 22nd 2022 (*GBIF.org, 2022*). We restricted our query to the data with spatial coordinates and which were inside the spatial extent of our rasters. A few observations were localized in the ocean near latitude 0° and longitude 0°. We assumed these were errors and removed all observations falling in the extent between latitudes 2°S and 2°N and longitudes 2°W and 2°E to keep only mainland sites. We did not use any additional geographical filters to retrieve as much data as possible. Being mindful of the recent and remarkable anthropogenic impact on African megafauna, we decided to restrict the occurrences used on the validation step to those recorded after the year 2000 (and, therefore, only records with date information). This decision was made after evaluating the overall temporal distribution of the GBIF records.

We then converted the occurrence data into raster format by determining which pixels had at least one GBIF occurrence. This allowed us to remove the effect of repeated sampling in some locations. These data were used to validate the areas identified as being ecologically unrealistic based on species interactions and occurrence data (see beginning of Methods section). To do so, we calculated the proportion of GBIF presence pixels occurring within both the original IUCN species range and the adjusted one (*i.e.,* the one without unrealistic food webs). We then compared these proportions for all predators to verify if the areas of data mismatch contained locations with GBIF observations, hence likely true habitats.

## Software

We performed all analyses using *Julia* v1.7.2 (*Bezanson et al., 2017*). We used the packages `SimpleSDMLayers.jl` (*Dansereau & Poisot, 2021*) to manipulate the raster layers, `EcologicalNetworks.jl` (*Poisot et al., 2019*) to construct and manipulate the interaction networks, and `GBIF.jl` (*Dansereau & Poisot, 2021*) to reconcile species names with the GBIF backbone taxonomy (*GBIF Secretariat, 2021*). We also used *GDAL* (*GDAL/OGR contributors, 2021*) to rasterize the IUCN range maps (initially available as shapefiles from the Spatial Data Download portal). All the scripts required to reproduce the analyses are available at https://doi.org/10.5281/zenodo.7374594.

## RESULTS

Mammal species found in the Serengeti food web are widespread in Africa, especially in grasslands and savannahs (Fig. 1A). From our analysis, most local networks (69.07%) built using the original IUCN range maps had at least one mammal species with a path to a primary producer (Fig. 1B), which reinforces that the interactions we observe in the Serengeti food web is representative of the interactions for these mammals in the whole African continent. On average, local food webs had almost half of their mammal species disconnected from basal species (mean = 46.2%, median = 33.3%). In addition, 16.6% of the networks only had disconnected mammals, and the number of mammal species varied from 1 to 28, with a mean of 6.7. As expected, the proportion of carnivores with a path to a primary producer was conditional on the total number of mammal species in each local network (Fig. 1C).

### Specialized predators have higher rates of range mismatch

If we consider that we cannot use areas where there are no superposition between predators and prey on ecological analyses, we lose more range area for predators with fewer prey (Fig. 2). For instance, both *Leptailurus serval* and *Canis mesomelas* have only one prey in the Serengeti food web (Table 1), each of them with a very small range compared to those of their predator. This discrepancy between range sizes promotes significant range loss. On the other hand, predators of the genus *Panthera* are some of the most connected species, and they also lose the least proportion of their ranges. This mismatch between predators and preys can also be a result of taxonomic disagreement between the geographical and ecological data. Although *Canis aureus* has the same number of prey as *Caracal caracal*, none of the prey taxa of the former occurs inside its original range (Table 1), which results in complete range loss.

There was a high variation in the overlap of predator and prey ranges (Fig. 3). The high density of points on the left-hand side of Fig. 3 indicates that most preys have small ranges in comparison to those of the set of carnivores in the networks, resulting in either low overlap between both ranges (bottom) or high overlap of ranges because much of that of the prey is within predators' range (top). The top-right side of the plot encompasses situations where the ranges of both predator and prey are similar and overlapping, while the bottom-right part of the plot represents a situation where the range of the predator is smaller than that of its prey and much of it occurs within the preys' range. For example, *Panthera pardus* had many preys occurring inside its range, with highly variable levels of overlap (Table 1). In general, species exhibited more consistent values of prey-predator overlap, than predator–prey overlap –indicated by the spread of points along the *x*-axis, yet more restricted variation on the *y*-axis (Fig. 3). There was also no overall relationship between the two metrics, or for any predator species.

### Validation with GBIF occurrences

The proportion of GBIF pixels (pixels with at least one GBIF occurrence) matching the IUCN ranges varied a lot for species with small ranges and way less for species with large ranges (Fig. 4, top). This means that species with large ranges had more area where their
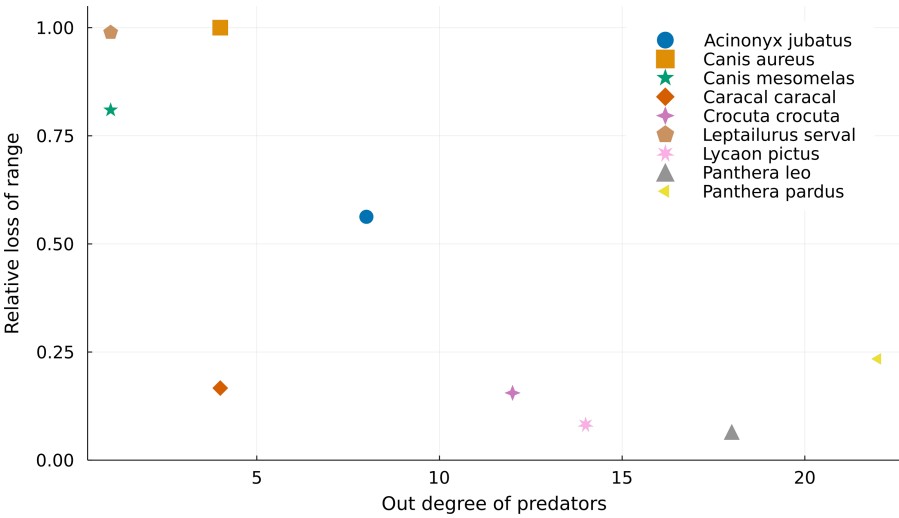

**Figure 2** **Relationship between the number of preys of each predator and their relative range loss.** Negative relationship between the out degree of predator species and their relative range mismatch. More specialized predators "lose" a higher proportion of their ranges due to mismatches with the ranges of their preys.

datasets for ecological and geographical information agreed. The lowest proportions of GBIF pixels occurred for species with small ranges. Amongst herbivores, *Rhabdomys pumilio* has a proportion of 25.6% of its presence pixels within its IUCN range, while predators have this proportion above 47% (such as *Lycaon pictus*, with 47.6%, and *Panthera leo*, with 49.3%). Nevertheless, some species with smaller ranges showed high data overlap (such as *Canis mesomelas*, with 94.1%, and many herbivores). Overall, predators and preys displayed similar overlap variations, and species with median and large ranges had higher proportions of agreement between GBIF, IUCN and interaction datasets.

The proportion of GBIF pixels in revised ranges can only be equal to or lower than that of the original ranges, as our analysis removes pixels from the original range and does not add new ones. Rather, the absence of a difference between the two types of ranges indicates that no pixels with GBIF observations, hence likely true habitats, were removed by our analysis. Here this proportion was mostly similar to that of the original IUCN ranges for most predator species (Fig. 4). Two species showed no difference in proportion (*Lycaon pictus* and *Panthera leo*) while four species showed only small differences (*Crocuta crocuta* lost 0.4% of the original data overlap; *Caracal caracal* lost 3.4%; *Acinonyx jubatus* and *Panthera pardus* lost 6.2%).

On the other hand, three species, *Canis aureus*, *Canis mesomelas*, and *Leptailurus serval* showed very high differences, with overlaps lowered by 100%, 58.4%, and 100% respectively. These last two species are also the only predators with a single prey in our metaweb. *Canis aureus* has four preys, but it has one of the smallest ranges in IUCN, which is not covered by any of its preys. This result reinforces the concern raised in the literature on the use of IUCN range maps for species that are not well known (*Herkt, Skidmore & Fahr, 2017*), demonstrating how small range species are likely to have their

**Table 1** **List of species analysed, their out and in degrees, total original range size (in pixels), and proportion of their ranges occupied by their preys and predators (values between 0 and 1).** Species are sorted according to the groups identified by *Baskerville et al. (2011)*. Notice how some species are isolated in the network (*Loxodonta africana*) and how *Canis aureus*'s range does not overlap with any of its preys.

| Species | Number of preys | Number of predators | Total range size | Proportion of range occupied by preys | Proportion of range occupied by predators |
|---|---|---|---|---|---|
| **Large carnivores** | | | | | |
| *Acinonyx jubatus* | 8 | 1 | 9,250 | 0.437 | 0.618 |
| *Crocuta crocuta* | 12 | 1 | 4,822 | 0.844 | 0.253 |
| *Lycaon pictus* | 14 | 0 | 427 | 0.918 | – |
| *Panthera leo* | 18 | 0 | 1,274 | 0.935 | – |
| *Panthera pardus* | 22 | 0 | 7,563 | 0.766 | – |
| **Small carnivores** | | | | | |
| *Canis aureus* | 4 | 1 | 816 | 0.000 | 0.782 |
| *Canis mesomelas* | 1 | 1 | 2,201 | 0.190 | 0.994 |
| *Caracal caracal* | 4 | 0 | 5,239 | 0.833 | – |
| *Leptailurus serval* | 1 | 1 | 4,319 | 0.011 | 0.978 |
| **Small herbivores** | | | | | |
| *Damaliscus lunatus* | 0 | 4 | 626 | – | 1 |
| *Hippopotamus amphibius* | 0 | 0 | 419 | – | – |
| *Kobus ellipsiprymnus* | 0 | 4 | 2,961 | – | 1 |
| *Ourebia ourebi* | 0 | 5 | 2,484 | – | 1 |
| *Pedetes capensis* | 0 | 2 | 1,318 | – | 1 |
| *Phacochoerus africanus* | 0 | 5 | 3,331 | – | 1 |
| *Redunca redunca* | 0 | 5 | 1,935 | – | 1 |
| *Rhabdomys pumilio* | 0 | 5 | 53 | – | 1 |
| *Tragelaphus oryx* | 0 | 2 | 2,316 | – | 0.990 |
| *Tragelaphus scriptus* | 0 | 3 | 3,999 | – | 0.985 |
| **Large grazers** | | | | | |
| *Aepyceros melampus* | 0 | 5 | 1,167 | – | 1 |
| *Alcelaphus buselaphus* | 0 | 4 | 2,307 | – | 1 |
| *Connochaetes taurinus* | 0 | 6 | 1,074 | – | 1 |
| *Equus quagga* | 0 | 5 | 786 | – | 1 |
| *Eudorcas thomsonii* | 0 | 6 | 51 | – | 1 |
| *Nanger granti* | 0 | 6 | 261 | – | 1 |
| **Hyraxes** | | | | | |
| *Heterohyrax brucei* | 0 | 1 | 1,961 | – | 0.973 |
| *Procavia capensis* | 0 | 1 | 5,312 | – | 0.647 |
| **Others** | | | | | |
| *Giraffa camelopardalis* | 0 | 1 | 607 | – | 0.473 |
| *Loxodonta africana* | 0 | 0 | 1,078 | – | – |
| *Madoqua kirkii* | 0 | 7 | 443 | – | 1 |
| *Papio anubis* | 0 | 1 | 2,571 | – | 0.937 |
| *Syncerus caffer* | 0 | 1 | 2,808 | – | 0.251 |
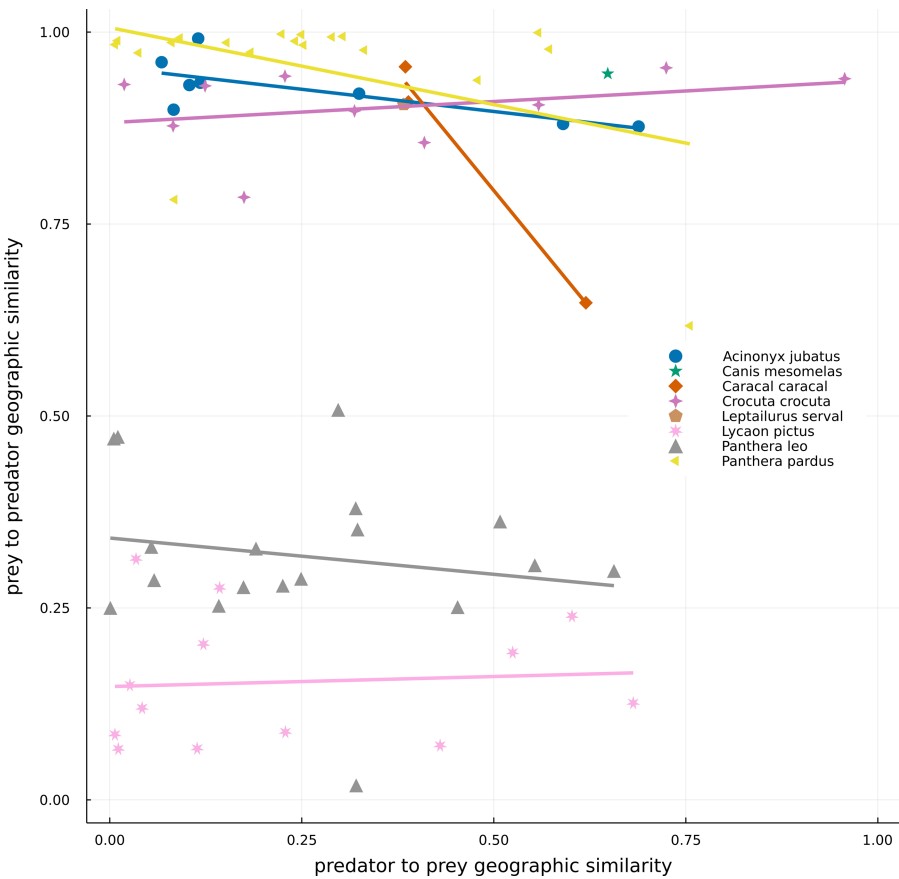

**Figure 3** **Geographical similarity between the original IUCN range maps of predators and preys.** Dots represent predator–prey pairs, with different symbols corresponding to different predators. For a given pair of species, the number $c$ of pixels where the focal species is present but not the other and the number $a$ of pixels where the predator and prey cooccur, were calculated. Geographic similarities were given by $a/(a+c)$, with the predator being the focal species in the predator to prey similarity ($x$-axis), while the prey is the focal one in the prey to predator similarity ($y$-axis). One of the predators, *Canis aureus*, is not represented in the image because it is an extreme case (where all its range is suppressed by the absence of preys) and it would make the interpretation of the data more difficult.

distribution underestimated in the IUCN database. Additionally, the fact that *Canis aureus* had such a conspicuous discrepancy between its original IUCN range and those of its preys, and between GBIF and IUCN data, may indicate a taxonomic incongruency between the three databases used here, which we explore in the Discussion section. Our results delineate how a mismatch between GBIF and IUCN databases differ greatly with small changes in herbivore species ranges, and it is somewhat positively related to range size for predator species. Moreover, we show that accounting for interactions does not necessarily aggravates this dissimilarity, but it is relevant for species about which we have little ecological information or for specialists groups.
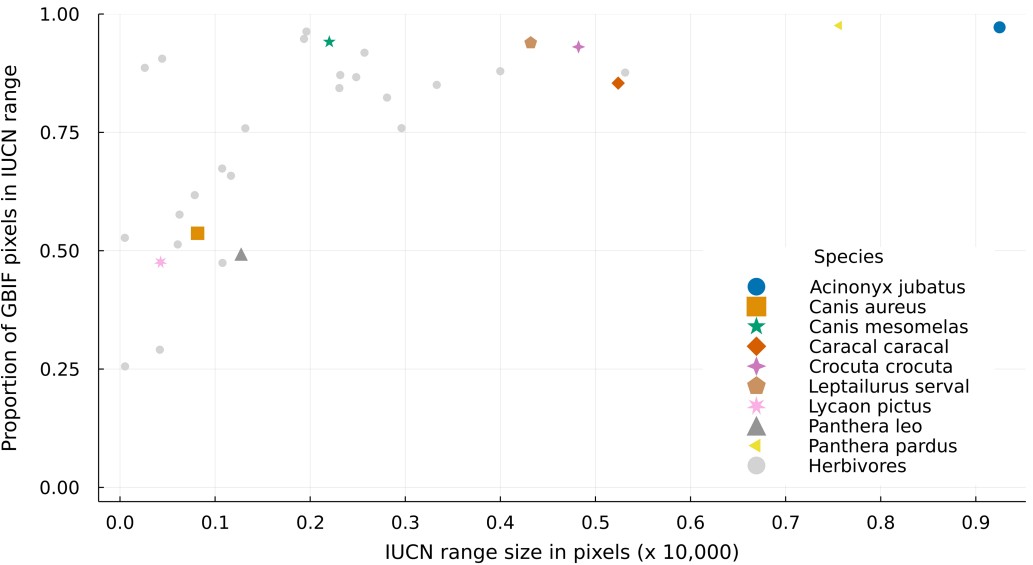

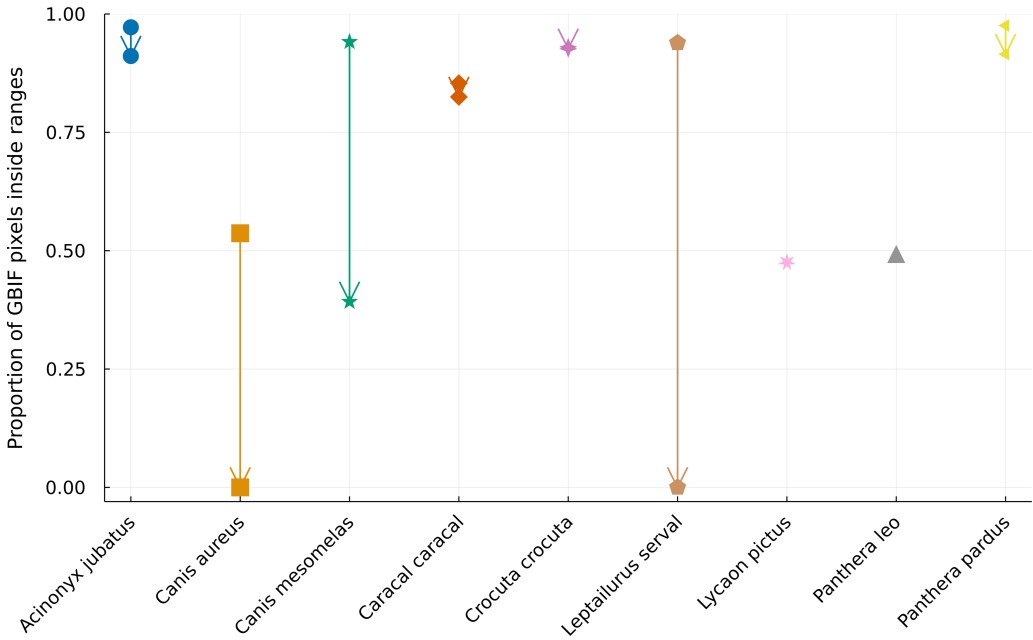

**Figure 4 Distribution of GBIF and IUCN mismatch between different range sizes.** Top panel: Distribution of the proportion of GBIF pixels (pixels with at least one occurrence in GBIF) superposed by the IUCN range data for different range sizes. Bottom panel: Differences between the proportion of GBIF pixels matching the original and cropped IUCN range maps for every predator species. Arrows go from the proportion inside the original range to the proportion inside the revised range, which can only be equal or lower. Overlapping markers indicate no difference between the types of layers. Species markers are the same on both figures, with predators presented in distinct colored markers and all herbivores grouped in a single grey marker. Pixels represent a resolution of 0.5 degrees.

## DISCUSSION

Here we identify areas of data mismatch between species range maps by using ecological interaction data (predator–prey interactions within food webs). Our results did
show a significant mismatch in the IUCN range areas of specialized and generalist predatory organisms and their prey, which highlights the importance of accounting for species interactions when estimating the range of a species. Although this type of data mismatch can be result of actual ecological processes, outdated occurrence data, taxonomic errors and more, we argue that, here, they rather indicate a lack of interaction sampling data.

The case of the golden jackal (*Canis aureus*) is a good illustration of how the taxonomic, geographical and ecological data can be used to validate one another. The jackal is a widespread taxon in northern Africa, Europe, and Australasia, generally well adapted to local conditions due to its largely varied diet (*Tsunoda & Saito, 2020*; *Krofel et al., 2021*). Because of that, we expected that the *Canis* species in our dataset would be the ones losing the least amount of range, with a higher value of the proportion of GBIF pixels within their IUCN range maps. However, the taxonomy of this group is a matter of intense discussion, as molecular and morphological data seem to disagree in the clustering of species and subspecies (*Krofel et al., 2021*; *Stoyanov, 2020*). This debate probably influenced our results: with originally only 64.9% of the GBIF pixels of the golden jackal overlapping with its IUCN data, we suspect that many of the GBIF occurrences refer to other *Canis* species, and that its taxonomic identification in the network database is probably outdated. This led to a complete exclusion of *Canis aureus* from its original range in our analysis, despite the fact that this species has four documented preys in our metaweb.

## Geographical mismatch and data availability

The lack of superposition between IUCN range maps and GBIF occurrences in our results suggests that we certainly miss geographical information about the distribution of either the prey or the predator. On the other hand, if both GBIF and IUCN occurrences tended to superpose and the species was still locally removed, this indicates that we don't have information about all its interactions (*e.g.*, predators may be feeding on different species than the ones in our dataset outside the Serengeti ecosystem). This rationale can be illustrated with three types of mismatches identified in our results.

First, *Panthera leo* was one of the species with no difference between ranges before and after our analysis, but 50.7% of its GBIF pixels did not superpose with the IUCN range (Fig. 4). In this particular case, the IUCN maps seem to agree with species interaction data. However, the disagreement between the IUCN and the GBIF databases is concerning and suggests that the IUCN maps might underestimate the lion's distribution.

On the other hand, *Leptailurus serval* and *Canis mesomelas* are two of the three species that have the higher proportion of mismatched range due to the lack of paths to a herbivore, but are also some of the species with the higher proportion of GBIF occurrences inside their original IUCN range maps (Fig. 4). This indicates that the information we are missing for these two species is related to either an additional interaction or to the presence of external interacting species. To illustrate that, we mapped the GBIF data for the prey of *Leptailurus serval*, with a mobility buffer around each point (Fig. 5). When considering GBIF data, approximately 36% of the prey's occurrences are within the portion of the predator's range that was divergent from its original IUCN data. With the buffer area, this
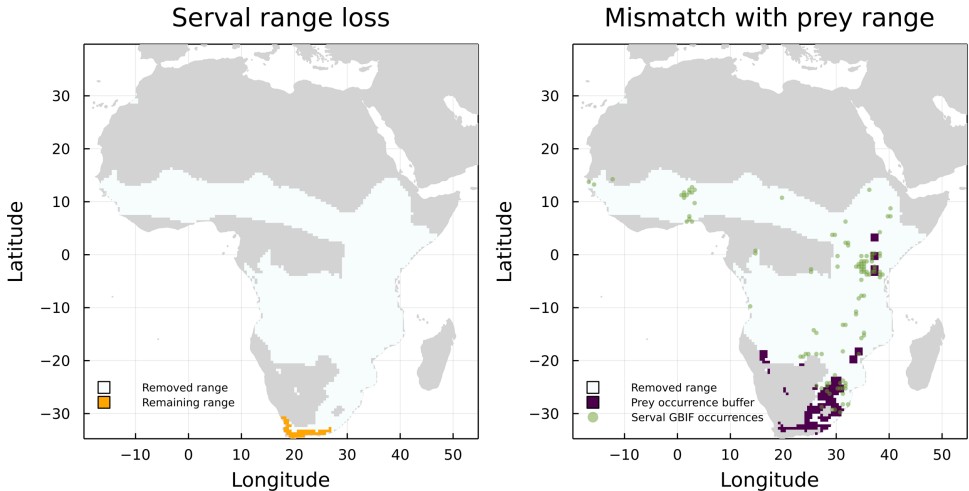

**Figure 5** **Comparison between the serval's IUCN range loss and its mismatch with GBIF data.** Mismatch between serval's range loss and GBIF occurrence of its prey. The left panel shows the reduction of serval's range when we consider the IUCN data on its prey. On the right panel, we added GBIF data on both serval and its prey, with a buffer for the prey to account for species mobility.

corresponds to 5.57% of the mismatched area. By adding GBIF information for the prey, we could therefore reduce the discrepancy of the range (or information) for the predator by 5.57% since its distribution is conditional on the occurrence of its preys. In other words, the range mismatch was exaggerated because we were missing information on the presence of an interacting species (*i.e.,* this also indicates that there is a mismatch—or complementarity—between the IUCN and GBIF data for their prey).

Finally, the extreme case of *Canis aureus* illustrates a lack of both geographical and ecological information: only half of its GBIF presence pixels and none of its preys occur inside its IUCN range. We believe, therefore, that the validation of species distribution based on ecological interaction is a relevant method that can further fill in information gaps. Nevertheless, it is imperative that more geographically explicit data about ecological networks and interactions become available. This would help clarify when cooccurrences can be translated into interactions (*Windsor et al., 2022*) and help the development of more advanced validation methods for occurrence data.

## Next steps

Here we demonstrated how we can detect areas of data deficit in species distribution data using ecological interactions. Knowing where questionable occurrence data are can be crucial in ecological modelling (*Hortal, 2008*; *Ladle & Hortal, 2013*), and accounting for these errors can improve model outputs by diminishing the error propagation (*Draper, 1995*). For instance, we believe our method is a way to account for ecological interactions in habitat suitability models without making the models more complex, but by making sure (not assuming) that the input data—the species occurrence—actually accounts for ecological interactions. Another application of this method is mapping areas where data are deficient, thus helping to identify priority sampling locations for interaction data, which

can, in turn, reduce uncertainty in network prediction. For example, if a certain pixel confirms the presence of a species both with IUCN and GBIF data, but lacks connection between species, this pixel has a high potential to hide an unobserved interaction and should therefore be a priority sampling location.

It is important to notice, however, that the quality and usefulness of this method are highly correlated with the amount and quality of data available about species' occurrences and interactions. With this article, we hope to add to the collective effort to decode the encrypted message that is the occurrence of a species in space and time. A promising avenue that adds to our method is the prediction of networks and interactions at large scales (*Strydom et al., 2021*; *Windsor et al., 2022*), for they can add valuable information about ecological interactions where they are missing. Additionally, in order to achieve a robust modelling framework towards actual species distribution models we should invest in efforts to collect and combine open data on species occurrence and interactions (*Windsor et al., 2022*), especially because we may be losing ecological interactions at least as fast as we are losing species (*Valiente-Banuet et al., 2015*).

## ACKNOWLEDGEMENTS

We acknowledge that this study was conducted on land within the traditional unceded territory of the Saint Lawrence Iroquoian, Anishinabewaki, Mohawk, Huron-Wendat, and Omámiwininiwak nations. We thank the editor and reviewers for their thoughtful comments, which considerably improved this manuscript.

### Funding

Gracielle Higino, Francis Banville, Gabriel Dansereau, and Norma Forero are funded by NSERC Computational Biodiversity Science and Services (BIOS$^2$) CREATE program; Francis Banville, Norma Forero, and Timothée Poisot are funded by Institute for Data Valorization (IVADO); Norma Forero and Timothée Poisot are funded by a donation from the Courtois Foundation; Gabriel Dansereau is funded by Natural Sciences and Engineering Research Council (NSERC) and the Fond de Recherche du Québec - Nature et Techonologie (FRQNT) doctoral scholarships; Timothée Poisot is funded by the Canadian Institute of Ecology & Evolution; Fredric Windsor is funded by the Royal Society (Grant number: CHL\R1\180156). The funders had no role in study design, data collection and analysis, decision to publish, or preparation of the manuscript.

### Grant Disclosures

The following grant information was disclosed by the authors:
NSERC Computational Biodiversity Science and Services (BIOS$^2$) CREATE program.
Institute for Data Valorization (IVADO).
The Courtois Foundation.
Natural Sciences and Engineering Research Council (NSERC).

The Fond de Recherche du Québec - Nature et Techonologie (FRQNT) doctoral scholarships.
The Canadian Institute of Ecology & Evolution.
The Royal Society: CHL\R1\180156.

## Competing Interests

The authors declare there are no competing interests.

## Author Contributions

- Gracielle T. Higino conceived and designed the experiments, performed the experiments, analyzed the data, prepared figures and/or tables, authored or reviewed drafts of the article, and approved the final draft.
- Francis Banville conceived and designed the experiments, performed the experiments, analyzed the data, prepared figures and/or tables, authored or reviewed drafts of the article, and approved the final draft.
- Gabriel Dansereau conceived and designed the experiments, performed the experiments, analyzed the data, prepared figures and/or tables, authored or reviewed drafts of the article, and approved the final draft.
- Norma Rocio Forero Muñoz conceived and designed the experiments, authored or reviewed drafts of the article, and approved the final draft.
- Fredric Windsor conceived and designed the experiments, performed the experiments, analyzed the data, prepared figures and/or tables, authored or reviewed drafts of the article, and approved the final draft.
- Timothée Poisot conceived and designed the experiments, authored or reviewed drafts of the article, and approved the final draft.

## Data Availability

The data used in this article is third-party and the code to retrieve them, as well as to produce the analysis, is available at Zenodo: Gabriel Dansereau, Gracielle Higino, Francis Banville, Fred Windsor, & Norma Forero. (2022). graciellehigino/ms_range_interactions: 2nd major revision (v2.0). Zenodo. https://doi.org/10.5281/zenodo.7374594.

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
