# Peer review of "Mismatch between IUCN range maps and species interactions data illustrated using the Serengeti food web"

_PeerJ, doi:10.7717/peerj.14620_

## Round 0.1 · original submission · Major Revisions

I was able to secure one review for this paper. The reviewer pointed out that some background is lacking in the introduction. Citing Gotelli, Graves & Rahbek 2010 PNAS in the intro could be useful. About the accuracy of range maps, see also Hurlbert & White 2005 Ecol Lett, Hurlbert & Jetz 2007 (these would be nice to see cited in 3rd paragraph of the introduction), but notice that the quality of the range maps also have a geographical context (e.g., Ficetola et al. 2014 J Biogeog). She also mentions that the section on geographical data validation needs improvement, by explaning how you produced absence data from presence-only data base. To which I would add the need to provide more details on how you cleaned GBIF data before actually using it. You also need to provide many more details on data retrieval, which date range ?
I tend to agree with her that the paper is indeed quite well written and helps placing trophic interactions in a macroecological context at the same time that it assesses the accuracy of IUCN's extent-of-occurrence polygons. Additionally, citation is lacking in many places throughout the text. For example, L. 68-70.
L. 2014-217 can be suppressed. Avoid citing tables and figures in the Discussion.
Finally, PeerJ uses a structured abstract. I highly recommend authors to adhere to it when resubmitting the revised version.

·

Basic reporting

This manuscript is very interesting and well-organized, with professional structure, figures, and tables, and very promising results. The writing is unambiguous, I just made some grammatical suggestions in the attached PDF. However, relevant prior literature must be referenced in your introduction to provide the necessary scientific basis for understanding. So, please provide more literature references in the first paragraph of your introduction. Below are other small tweaks to improve your manuscript:

Lines 7-9: Please provide a short description for each example (taxonomic errors, geographic inaccuracy, or sampling biases).

Lines 9-11: Please cite studies that have done this before! If there are no previous studies, it is important to highlight the novelty of your study.

Lines 21-23: Are the cited literature examples of models that do not take ecological interactions into account? Use e.g. before.

Experimental design

This research is within the Aims and Scope of the PeerJ journal. The research question is very well defined and relevant. It can help identify and fill a very important knowledge gap about species distribution. Moreover, the applied methods are very well described, except for data validation (more details below). Here I also highlighted some suggestions to improve your manuscript:

Lines 68-70: This mismatch cannot just be a result of the overestimation of the predator’s range but it could be the result of misestimating both the predator's and prey's range. Please rephrase the sentence.

Lines 124-131: I have a doubt about your data validation. First, you mentioned that you compiled point observation data from GBIF. Then you converted that data in the presence or absence of the focal taxon. How did you transform these data in the presence or absence once the GBIF provides just occurrence points (i.e. presence data)? This was not well explained. In general, absence data are not provided because the non-detection of a species in a location does not mean a true absence. This could just be a detection issue. Thus, it is needed to calculate the pseudo-absence data. Please explain this better in your manuscript.

Validity of the findings

No comment

Additional comments

Although the manuscript is well written, detailed, and with very important results, I suggested that the article be accepted with major revisions because of the data validation step, which was not clear to me. As this step implies in the results, I suggest further details for publication.

---

## Round 0.2 · Major Revisions

I have now consulted a new reviewer who is specialist on large scale biodiversity patterns of Africa. So, while one reviewer from the previous round is satisfied by your responses to his/her comments, I believe there's still some room for improvement in this manuscript. I really believe this could be a potentially good and impactful contribution, but authors need to do a better job in addressing those issues raised at this time.

·

Basic reporting

No comment

Experimental design

No comment

Validity of the findings

No comment

Additional comments

This revision is much improved. I appreciate the answers about the data validation and pseudo-absence data because now it is so clearer for me. In general, the critical comments are correctly addressed, and this paper is much more convincing now. Thank you.

Reviewer 2 ·

Basic reporting

In this study, Higino et al., analyze the uncertainty of IUCN ranges of 41 species (23 herbivores and 9 carnivores) using GBIF records and the Serengeti food web dataset (Baskerville et al. 2011). The Manuscript is well written and clear. I like the idea but there are a number of issues that are not clear.

Experimental design

1. In the extent of the study, the authors mention that they “restricted the rasters a spatial extent comprised between latitudes 35°S and 40°N and longitudes 20°W and 55°E. “ – which is the whole of Africa. Since the study uses a dataset from Tanzania (Serengeti food web dataset), roughly 3% of the continent, how did the authors deal with the presence of putative prey items in other parts of the continent that are absent from the Serengeti food web dataset? For example Papio ursinus, Sylvicapra caffra and S. grimmia don't occur in Tanzania but are prey items to many carnivores in Africa.
2. The resolution: it is well studied that IUCN range maps should not be converted to grids smaller than 50km due to commission errors. This becomes a major problem when dealing with widespread species such as the ones used in this study: African large and medium mammals. See https://doi.org/10.1111/1365-2664.12771 and https://doi.org/10.1073/pnas.070446910
3. Gbif records: using records from any period in time from gbif will include historical relics from extinct populations. Many large and medium mammals, especially large carnivores have suffered major declines in the last decades in Africa. How did the authors deal with the fact that records from a species recorded 50 years ago may represent extinct populations?
4. The gbif records for large mammals are especially problematic. Did the authors include observations as well? iNaturalist for instance now feeds data directly to GBIF. Having museum specimens of lions and other large/medium mammals are especially problematic and difficult to achieve nowadays. Many countries such as Malawi and Mozambique conduct very little research and many areas are not documented, not even on GBIF. Therefore the occurrences we see on GBIF are biased towards countries that have available data. How did the authors account for this? See https://doi.org/10.1093/sysbio/syaa090
5. Taxonomic groups: Carnivores feed on a diversity of items, beyond herbivores. Their diet might include birds, fish, reptiles, amphibians, and even invertebrates. How did the authors account for the fact that despite missing the herbivores from Serengeti, there were other sources of protein that could sustain the species in the continent?

The discussion also needs work to add context to the actual results from the study. As for now, it could easily be a continuation of the introduction. The same holds true for the first section of the methods.

Validity of the findings

Validity of the findings: As it stands, and without explaining how the above mentioned issues were treated I have little trust in the findings of this manuscript.

---

## Round 0.3 · accepted · Accept

Thank you for making minor, but relevant, adjustments to the manuscript. I’m more than happy with this version and therefore consider it ready to be accepted.